# The Use of Surface-Modified Nanocrystalline Cellulose Integrated Membranes to Remove Drugs from Waste Water and as Polymers to Clean Oil Sands Tailings Ponds

**DOI:** 10.3390/polym13223899

**Published:** 2021-11-11

**Authors:** John Jackson, Ali Moallemi, Mu Chiao, David Plackett

**Affiliations:** 1Faculty of Pharmaceutical Sciences, University of British Columbia, 2405 Wesbrook Mall, Vancouver, BC V6T 1Z3, Canada; davidplackett@gmail.com; 2Department of Biomedical Engineering, University of British Columbia, Vancouver, BC V6T 1Z3, Canada; ali.moallemi91@gmail.com (A.M.); muchiao@mech.ubc.ca (M.C.)

**Keywords:** water treatment, nanocrystalline cellulose, membrane filtration, cationic polymer, drugs, oil sands

## Abstract

There is an urgent environmental need to remediate waste water. In this study, the use of surface-modified nanocrystalline cellulose (CNC) to remove polluting drugs or chemicals from waste water and oil sands tailing ponds has been investigated. CNC was modified by either surface adsorbing cationic or hydrophobic species or by covalent methods and integrated into membrane water filters. The removal of either diclofenac or estradiol from water was studied. Similar non-covalently modified CNC materials were used to flocculate clays from water or to bind naphthenic acids which are contaminants in tailing ponds. Estradiol bound well to hydrophobically modified CNC membrane filter systems. Similarly, diclofenac (anionic drug) bound well to covalently cationically modified CNC membranes. Non-covalent modified CNC effectively flocculated clay particles in water and bound two naphthenic acid chemicals (negatively charged and hydrophobic). Modified CNC integrated into water filter membranes may remove drugs from waste or drinking water and contaminants from tailing ponds water. Furthermore, the ability of modified CNC to flocculate clays particles and bind naphthenic acids may allow for the addition of modified CNC directly to tailing ponds to remove both contaminants. CNC offers an environmentally friendly, easily transportable and disposable novel material for water remediation purposes.

## 1. Introduction

There is an increasing global awareness of the need to clean up waste water. Contaminants range from micropollutants such as household-discarded drugs to huge tailings ponds next to oil sand bitumen extraction facilities. Whilst drugs are only present at very low concentrations, they may have pronounced cumulative effects for both aquatic life and humans. According to the World Health Organization (WHO), nanogram-level concentrations of pharmaceuticals in drinking water raised concerns regarding the potential risks to human health [1,2,3]. The potential environmental impact of huge volumes of contaminated water in tailing ponds is obvious. Current methods to remove drugs from water or to remediate tailings ponds are ineffective. There is an unmet need for systems that themselves do not create alternative problems like high energy use, fouling, non-degradable clean up materials, transportation or intense maintenance issues.

The pollution of water is a major environmental issue. For systems that adsorb and remove pollutants, it is preferable for the adsorbate to have high surface areas to maximize direct adsorption, and to be inexpensive and biodegradable. Considerable attention has been paid to the potential use of nanomaterials in environmental remediation science due to the high surface area to volume ratio, easily functionalized surfaces for specific trapping and easy transport [4,5]. Cellulose, and in particular nanocellulose, is such a material and has the added advantage of being easily surface derivatized to create surfaces for specific adsorbing purposes [6].

Nanocrystalline cellulose (CNC) is extracted from wood biomass by acidic extraction methods producing nanosized cellulose rods with a very high surface to volume ratio. Due to this large surface area, high aspect ratio, and strength, CNC is being investigated as a nano-composite material. A thorough review of the chemical properties of CNC is available from Habibi et al. [7]. Nanocrystalline cellulose, either native or surface modified, has been proposed for use in removing metal, oil and pharmaceutical contaminants from water [8,9,10,11]. Because of the ability to bind drugs, CNC has also been used in formulations for drug delivery underlining the potential for the material to sequester drugs from polluted waters [12,13].

On a micropollutant front, there are numerous types of drugs disposed of everyday down household sewerage systems. Most drugs are degraded in the activated sludge phase of sewage treatment [14,15]. However, some drugs are only partially degraded and have significant biological activity. These include the heavily prescribed and disposed of group called non-steroidal anti-inflammatories (like ibuprofen, naproxen and diclofenac) and the hormones based on estrogen [15,16,17]. Indeed, diclofenac and estradiol are known to be the most important disposed drugs on the EU 2015 watch list for pharmaceutical contaminants [16]. The sewage sludge degradation of estradiol is poor and diclofenac persists at high levels after sewage treatment [15,16]. These agents are harmful to numerous biological species at low concentrations and it should be remembered that in many water-restricted countries, household drinking water is recycled and provided back to household water supplies raising the risk of cumulative exposure to low drug concentrations and associated toxicity. There exists a need to remove undegraded drugs like estradiol and diclofenac after the activated sludge process (before efflux into rivers or the ocean) or to treat recycled drinking water before release from household taps. In terms of macropollutants, the process of extracting bitumen from the Alberta oil sands requires the use of large amounts of heated water and generates huge volumes of polluted waste water which are pumped into tailings ponds. These liquids contain sand, clay, residual bitumen and toxic complex hydrocarbons (mostly naphthenic acids). Following a rapid sedimentation of large particulates like sand, the residual clay and bitumen do not settle quickly and create an escalating environmental problem centered on the inability to reclaim the water and land. By 2015, less than 2% of tailings ponds had been reclaimed with untreated tailings ponds taking decades to clear [18].

Despite extensive research no single simple method yet exists to address these problems which are projected to grow as oil extraction expands over the next 50 years. Most efforts focus on the use of polymer flocculants to disrupt the repulsive forces between clay particles that keep them in suspension [19,20,21]. However, the composition of tailings clays and their suspension in the tailings water is complex, highlighted by the presence of various clay types, their ability to swell and hold water and potential interactions of clays with bitumen which may mask their surface charge and flocculation potential. Many clay particles are negatively charged at neutral pH and become even more negative at high pH such as tailings ponds at pH of approximately 8.5 [19]. The widely -used polymer Percol 727 (a high molecular weight polyacrylamide (PAM)) is ineffective at flocculating smaller clay particles such as montmorillonites which have very high specific surface areas and massive cation exchange capacities [19,20,21,22]. PAM-flocculated tailings liquids still may have 3% residual amounts of suspended clays where the limit for water disposal is 0.5% making further treatments necessary to remove these highly negatively charged species.

The is an unmet need for an inexpensive cationic material capable of flocculating and removing these small negatively charged clays like montmorillonites. Other workers have had limited success in flocculating kaolin clays using cellulose [23,24]. In these studies, the use of cationically modified nanocellulose (CTAC-CNC) to flocculate Cloisite (a commercial montmorillonite) clay at pH 8.5 was investigated and an effective flocculation of clay with this agent was observed.

Aliphatic and aromatic naphthenic acids are the principal small molecule contaminants in tailings ponds [25,26,27,28,29]. These agents are corrosive to all engineering equipment and highly toxic to aquatic life and potentially to humans if they reach high concentrations in rivers [30,31,32]. Naphthenic acids do not degrade easily by natural methods and accelerated oxidation using ozone or oxygen radical forming agents are only partially successful [30]. A variety of methods have been investigated for the removal of these chemicals with limited success [25,26,27,28,29,30].

Nanocrystalline cellulose (CNC) is the smallest extractable subunit of cellulose from inexpensive and abundant wood biomass. CNC has the benefit of a negative surface charge and a huge surface to volume ratio, allowing for high levels of surface modification. Previous studies have demonstrated that both chemically modified and unmodified CNC may bind various drugs with the intention of using the biocompatible CNC as a controlled drug release system [12]. The ability of CNC to bind and hold either charged or hydrophobic drugs hints at their potential to be used to remove unwanted chemical species (like drugs) from waste water. The overall objective of this study was to investigate the use of either cationic or hydrophobically modified CNC for various environmentally significant purposes. Specifically, these purposes include removing drugs from waste water and removing clays and organic contaminants from oil sands tailings ponds. Accordingly, CNC was integrated into membrane filters to remove estradiol or diclofenac from water and the modified CNC materials were used to flocculate clays and bind naphthenic acids. The novelty of these studies lies in the potential use of an inexpensive, ecologically compatible material (CNC) as an easy to use material for environmental remediation purposes.

## 2. Materials

CNC was provided by Alberta Innovates (Edmonton, AB, Canada) as a suspension in water at 1.8% *w*/*v*. All solvents were HPLC grade and obtained from Thermo Fisher Scientific (Ottawa, ON, Canada). Millipore filtration centrifuge tubes (0.22 um) were obtained from Millipore Canada (Oakville, ON, Canada). Large filtration units and filters (McMaster-Carr code 1354 N12 0.22 µm) were obtained from McMaster-Carr (Douglasville, GA, USA). Cloisite (Na+) clay samples were supplied by Southern Clays (Gonzales, TX, USA). All plasticware was supplied by Sarstedt (St Leonard, QC, Canada). All other reagents including the drugs estradiol (beta-estradiol) and diclofenac, naphthenic acids (4-(-2-naphthyloxy) butanoic acid and 4-(4-tert butyl phenyl) butanoic acid), conjugation reagents, etc. were purchased from Sigma Aldrich (Oakville, ON, Canada).

## 3. Methods

CNC was covalently modified using dioctenyldimethyl succinic anhydride (DDSA) (hydrophobic surface) or using polydiallyl dimethyl ammonium chloride (PDMC) (cationic surface).

### 3.1. Dioctenyldimethyl Succinic Anhydride-Nanocrystalline Cellulose (DDSA-CNC) Synthesis

A sample of CNC (2 g) was dispersed into 50 mL of distilled water and then centrifuged at 3500 rpm for 10 min. After removing the supernatant liquid, acetone (30 mL) was added, and the mixture centrifuged before the supernatant acetone was discarded. This process was then repeated twice. After discarding the acetone, the CNC was solvent-exchanged into *N*,*N*-dimethylformamide (DMF) by adding DMF (30 mL) into each of two centrifuge tubes. After shaking for 5 min, 20 mL of the mixture was transferred into each of two further centrifuge tubes. Then, 20 mL of acetone was added to each tube and the tubes were then centrifuged at 3000 rpm for 10 min before discarding the supernatant liquid. In the next step, DMF (50 mL) was added and mixed before the supernatant was poured off into a beaker and heated at 85 °C for about 15 min to ensure acetone evaporation. Then, dimethylaminopyridine (DMAP) (0.5 g) was added. The DDSA reactant (2 g) was mixed with 10 mL DMF and this solution was added to the CNC dispersion containing DMAP. After 4 h at room temperature, the dispersion was transferred to a filter flask with 140 mL acetone. A sample of this dispersion (40 mL) was centrifuged at 3400 rpm for 5 min before discarding the supernatant. This process was repeated twice before the supernatant was discarded and the modified CNC allowed to air dry for 24 h. This method is similar to that of Leszczyńska et al. [33].

### 3.2. Polydiallyl Dimethyl Ammonium Chloride (PDMC)-CNC Synthesis

CNC (2 g) was dispersed in 100 mL of distilled water and 100 mL of a 20% *w*/*v* PDMC solution was then added with stirring for 24 h. In the next step sodium chloride (20 g) was stirred into the CNC suspension for a further 24 h. After centrifugation at 620 rpm for 1 h, the supernatant was discarded. The suspension was then washed with distilled water by way of centrifugation, which was repeated five times and the modified CNC sediment was then collected. This sediment was freeze-dried for 24 h to yield a final PDMC-CNC product. This method is similar to that of Huang et al. [34]

### 3.3. Preparation of Cationic CNC (CTAC-CNC) or Hydrophobic CNC (CTAB-CNC) (Both Non-Covalent Methods)

Cationic CNC (CTAC-CTAC-CNC) was made by incubating 200 mL of CNC (18 mg/mL water) with 3-chloro hydroxy propyl trimethyl ammonium chloride (CTAC) in the presence of excess sodium hydroxide for 12 h. The mixture was centrifuged at 2000× *g* in a Beckman table top swinging bucket centrifuge for 30 min and the CNC pellet was resuspended and washed four times with centrifugation and made up to a final concentration of 15 mg/mL in water. Hydrophobic CNC was manufactured by incubating 200 mL of CNC (18 mg/mL) with cetyl trimethyl ammonium bromide (CTAB) for 30 min followed by centrifugation and washing. For studies with estradiol, CTAB was used at 10 mM and for naphthenic acid studies CTAB was used at either 7.5 or 15 mM. For estradiol binding studies the CTAB-CNC was used at 15 mg/mL and for naphthenic acid binding studies it was used at either 7.5 mg/mL or 15 mg/mL.

### 3.4. Drug or Contaminant Analysis

Diclofenac was measured using UV/VIS spectrophotometric methods at 275 nm. Briefly this method involves measuring the absorbance of light at 275 nm using a Cary 50 spectrometer (Agilent, Santa Clara, CA, USA). A set of calibration standards in the 0–10 µg/mL range was initially measured to create a linear standard curve. and estradiol was measured using high-performance liquid chromatography (Waters Acquity system) with a mobile phase of 50% water: 50% acetonitrile a flow rate of 1 mL per minute and detection at 280 nm as previously described [35]. Both drugs gave linear calibration curves in the 0–10 µg/mL range with correlation coefficients of 0.99. The naphthenic acid contaminants were analyzed using UV/VIS methods with detection at 362 nm. Calibration curves were linear in the 0–50 µg/mL range with correlation coefficients great than 0.985.

### 3.5. Filtration

For small-scale membrane filtration, 50 mL centrifuge tubes with 0.22 µm Millipore Steriflip filters (Oakville, ON, Canada) were used as shown in Figure 1a. We suspended 5, 10 or 20 mg of derivatized CNC in 5–10 mL of water and dispersed it using an ultrasonic tip disruptor sufficient to fully disperse the material without heating. The nanosuspension was then poured onto the filter and vacuum applied until all the material had filtered. The membrane was then washed three times using 5 mL of water under a vacuum and all the filtered liquid was discarded. The filter was then ready for application of a drug solution. Measured using optical density at 300 nm, this method removed 100% of CNC from the water so that it was bound to the membrane filter. Five mL of drug solution (2 µg/mL estradiol or diclofenac) was then added to the membrane and the solution was filtered through under vacuum. The filtered solution was then saved for drug analysis and the amount bound was calculated by subtracting the amount of drug in the filtered solution from the original drug amount in solution applied to the membrane filter.

For larger-scale filtration, 2 L capacity systems were used as shown in Figure 1b. Initially a 1 µm filter was tried in a household “Rainfresh”^TM^ under the counter inexpensive system. However, when CNC was filtered through only 40% was bound to the filter as measured by the reduction in optical density at 300 nm. A more expensive system (McMaster-Carr code 1354 N12 polypropylene plastic filter) with a 0.2 μm filter was then used and this removed 100% of the CNC from the water. Two grams of CNC was dispersed in water using an ultrasonic tip disruptor and made up to 2 L of water. Both the interior and exterior sections of the filter were filled with water. This solution was filtered through the apparatus under vacuum and then vacuum washed three times with 2 L of water.

### 3.6. Loading of CNC on to Small Scale Filters

Using unmodified CNC at 1 mg/mL, 5 mL was sequentially filtered through the small filter and the time required for full filtration was recorded.

### 3.7. Zeta Potential Measurments

Zeta potential was measured using a Zetasizer instrument (Malvern Instruments, Malvern, Worcester, England)

### 3.8. Drug Binding Studies

For binding of diclofenac, the PDMC-CNC was loaded onto the small-scale filter either at 5 mg or 20 mg total per filter. We passed 5 mL of diclofenac at 2 µg/mL through the filter and the amount of drug in the filtered liquid (unbound fraction) was analyzed and discarded. The process was repeated. For large-scale filtration 2 g of the PDMC-modified CNC was loaded onto the filter. We filtered 2 L of diclofenac at 2 µg/mL through the system under vacuum and the filtered liquid was analyzed for unbound drug content.

For estradiol binding with CTAB-CNC, the CTAB was used at 10 mM for modification and 25 mg of CTAB-CNC was integrated into the small-scale membrane and repeated 1 mL volumes of estradiol at 2 µg/mL were filtered. In a second experiment only 5 mg of CTAB-CNC was integrated into the membrane and larger volumes (5 mL) of estradiol (2 µg/mL) were filtered.

The DDSA-CNC was loaded onto the small filter at 5 mg or 10 mg total per filter and at 2 g onto the large-scale filter. Five mL (small scale) or 2 L (large scale) of estradiol at 2 µg/mL were filtered through the filters and the amount of unbound drug analyzed by HPLC methods as described in the Methods section.

### 3.9. Clay Flocculation Experiments

Cloisite clay was suspended in water at 10 mg/mL using ultrasonic tip sonication for 2 min without heating. This process was repeated twice. The resulting suspension only sedimented if the centrifuge force rose above 600 rpm (approx. 70× *g*). Two mL of the clay suspension was added to a 5 mL tube followed by various volumes of the CTAC-CNC suspension at 15 mg/mL. These volumes ranged from 25 to 600 μL and all tubes were topped up with water to a final volume of 2.6 mL. Tubes were shaken and left to stand for 5 min followed by analysis of the optical density (A 500 nm) of the top 0.7 mL in a UV/VIS spectrophotometer.

### 3.10. Naphthenic Acid Binding to CNC

Naphthyloxy butanoic acid or butyl phenyl butanoic acid were dissolved in water (pH adjusted to 8.3 using very dilute sodium hydroxide) at 125 µg/mL and serial diluted. CTAB- or CTAC-modified CNC was added at either 7.5 or 15 mg/mL. For CTAB experiments, sodium chloride at 10 mM was added to facilitate sedimentation under centrifugation. After 60 min, the CNC was pelleted by high-speed centrifugation and the unbound fraction of the naphthenic acids measured by UV/VIS spectroscopy.

## 4. Results

### 4.1. Filtration Methods

The apparatus for small- and large-scale membrane filtration is shown separately in Figure 1 and Figure 2. All forms of CNC were captured on the 0.22 µm small-scale Millipore filter as observed by the optical density (A500 nm) of the filtrate solution which was zero and clear. It was possible to filter numerous CNC suspension volumes (each containing 5 mg weights of CNC) through the filter with the accumulation of CNC on the filter having little effect on the filtration time, which was about 3 min for 5 mL of suspension (1 mg/mL) as shown in Figure 3. Even after 10 filtrations and 50 mg of CNC the membrane was unclogged and still able to integrate more CNC.

### 4.2. Binding of Estradiol to CTAB-CNC Integrated Small-Scale Membranes

When 25 mg of CTAB-CNC was integrated into the filter there were high levels of estradiol removal from water, as seen by the high levels of binding in Figure 4. Binding refers to the removal of a certain percentage of the applied drug. The % drug bound decreased from almost 100% on the first filtration run to a little under 60% bound after 22 repeated filtrations as seen in Figure 4a. The decreased binding with each run was approximately the same over the course of the 22 filtrations (approximately 2% decrease per filtration run). When only 5 mg of CTAB-CNC was loaded into the membrane filter and 5 mL volumes of estradiol at 2 µg/mL were filtered slightly lower levels of drug binding were observed as seen in Figure 4b. Although almost 100% of the drug was removed on the first filtration run, the % drug bound on each subsequent run dropped by approximately 16% for each of the next 5 runs with less than 20% of the loaded drug removed on each run after that. Using 20 repeated filtrations of 1 mL volumes of drug solution (total of 40 μg of drug applied to filter) to filters containing 25 mg of CTAB-CNC, the 60% drug binding level was maintained. However, for filters containing 5 mg CTAB-CNC after four filtrations of 5 mL (total 40 μg of drug) there was less drug binding (50% binding). At the end of estradiol filtration experiments, the addition and filtration of 5 mL of ethanol through the drug-adsorbed CNC-integrated membrane allowed for 100% recovery of the estradiol.

### 4.3. Zeta Potential of Covalently Modified CNC

Zeta potential measures the surface charge on solid surfaces and gives an indication of whether the charged drugs will bind to such surfaces. The zeta potential of unmodified CNC was −50 mV whereas covalent modification by cationic PDMC resulted in a positive charge across a range of pH as shown in Figure 5. The positive charge on the CNC was pH dependent and dropped with increased pH. This pH dependency results from protonation of the covalently bound species at lower pH values further confirming an effective conjugation process. All surface charges remain strongly positive at all pH values and are likely to provide good binding sites for negatively charged drugs like diclofenac, especially in any reasonably anticipated water pH (perhaps 5 to 9 where the zeta potential only changes from approximately 60 to 40). The zeta potential of DDSA covalently modified CNC was reduced to −20 mV with the same value recorded at all pH values.

### 4.4. Binding of Estradiol to DDSA-CNC Integrated Small Scale Membranes

The % binding of estradiol to either 5 mg or 10 mg DDSA-CNC loaded filters was found to be weak as seen in Figure 6a,b. Using 5 mg of DDSA-CNC resulted in just 50% of available drug being bound to the membrane and this binding dropped off quickly to almost zero by run 3. Similarly, the use of 10 mg of DDSA-CNC on the small filter resulted in more drug being bound than for 5 mg loaded membranes with over 70% available drug bound on the first run dropping to approximately 30% for runs 3 to 6 with a drop off to zero % binding by run 9.

### 4.5. Binding of Estradiol to DDSA-CNC Integrated Large Commercial Filter Membranes

When DDSA-CNC (2 g) was integrated into a large commercial filter the binding of estradiol (2 L of solution at 2 µg/mL) was significantly higher with 67% of available drug bound on the first run (Figure 7). However, although this dropped slightly after the first filtration run the binding stabilized at approximately 50% levels for all subsequent runs (12 completed runs). The reason for the slightly lower values between run 2 and 5 is not known.

### 4.6. Binding of Diclofenac to PDMC Integrated Small Scale Membranes

The binding of diclofenac to either 5 mg or 20 mg PDMC-CNC loaded membranes was very high as seen in Figure 8a,b. For 5 mg PDMC-CNC, initial binding levels were at 90% then stabilized slightly lower at between 70% and 90% for 40 repeated filtration runs (Figure 8a). Using 20 mg PDMC-CNC, slightly higher levels of binding were observed and started at 90% binding on the first run dropping and stabilizing at between 80% and 90% after run 6 (12 runs completed) (Figure 8b).

### 4.7. Binding of Diclofenac to PDMC Integrated Large Commercial Filter Membranes

The binding of diclofenac to large-scale membranes was lower at between 47% and 49% for six runs dropping and stabilizing at a level of approximately 40% after that (17 filtration runs completed) as seen in Figure 9.

### 4.8. Oil Sands Treatments and Clay Flocculation

Cloisite clay had a negative zeta potential of −40 mV as seen in Table 1. The charge on unmodified CNC was also negative at −50 mV. However, this dropped to −20 mV for CTAB modification and became highly positive for cationically modified CTAC-CNC (+41 mV) (Table 1).

The addition of CTAC-CNC to Cloisite clay suspensions resulted in a concentration dependent binding and flocculation effect as seen in Figure 10. Flocculation levels were approximately 10% for each mg of CTAC-CNC added so that after 9 mg almost 100% flocculation had occurred.

### 4.9. Binding of Naphthenic Acids to either CTAC-CNC or CTAB-Modified CNC

The binding of both naphthenic acids to CTAC-CNC was weak as shown in Figure 11A. Binding levels of between 55% and 40% were observed over a range of butyl phenyl butanoic acid concentrations and approximately 30% binding levels for naphthyloxy butanoic acid (Figure 11A). The binding of both naphthenic acids to CTAB-CNC was strong across the full range of naphthenic acid concentrations (Figure 11B,C). Using CTAB modification of CNC at 15 mM and CTAB-CNC at 15 mg/mL, there was approximately 90% binding of naphthyloxy butanoic acid dropping to approximately 40% using CTAB (7.5 mM modified) bound to CNC at 7.5 mg/mL and slightly higher at between 40% to 60% for CTAB (7.5 mM modified) bound to CNC at 15 mg/mL (Figure 11B). These values were unaffected by naphthenic acid concentration.

The binding of butyl phenyl butanoic acid to CTAB-CNC (Figure 11C) followed a similar pattern to naphthyloxy butanoic acid binding data except with values slightly higher. Using CTAB modification of CNC at 15 mM and CTAB-CNC at 15 mg/mL there was approximately 90% binding, dropping to approximately 60% using CTAB (7.5 mM modified) bound to CNC at 7.5 mg/mL, and slightly higher at between 65% to 90% for CTAB (7.5 mM modified) bound to CNC at 15 mg/mL (Figure 11C). These values were unaffected by naphthenic acid concentration.

## 5. Discussion

In this study, the potential use of surface-modified CNC to remove estradiol or diclofenac from water and the ability of modified CNC to reduce the two major environmental polluting aspects of the oil sands industry, namely the persistent clay and naphthenic acid content of tailings ponds, was investigated. Nanocrystalline cellulose is a readily available, ecologically suitable material for remediation purposes and it has the added advantage of being easily surface derivatized to create surfaces for specific adsorbing purposes [36,37,38]. Nanocrystalline cellulose, either native or surface modified, has been proposed for use in removing metal, oil and pharmaceutical contaminants from water [38].

Initially, CNC was found to partially pass though large pore size (1 μm) filters but was reproducibly and fully trapped on 0.2 μm filters without blocking water passage (Figure 3) establishing the concept of a simple one-step procedure for modifying existing commercial filters for the purpose of drug removal. In this study, known methods to surface modify CNC into hydrophobic surfaces in an attempt to bind estradiol or naphthenic acids were used. Similarly, cationic surfaces were provided to bind negatively charged diclofenac or naphthenic acids. These methods included simple chemical species binding to CNC or more complex covalent modification methods. The zeta potential of native CNC was found to be negative at −50 mV (Table 1) as previously reported [12]. It has been previously shown that the hydrophobic modifying group CTAB bound well to CNC as reported by a less negative zeta potential [12] and this was confirmed in this study. A similar change in the zeta potential of CNC covalently modified with the hydrophobic DDSA group was also found and this zeta potential was found to be −20 mV over a wide range of pHs (Figure 5). The efficient adsorption of cationic CTAC to CNC was confirmed by the positive zeta potential of +41 mV (Table 1). Similarly, the effective covalent cationization of CNC by PDMC was also confirmed by a positive zeta potential over a wide range of pH with the reduction of positive zeta potential at higher pH (Figure 5) reflecting the mild acidic nature of the cationic ammonium ion.

The two important aspects of removing discarded drugs from water are removal from waste water treatment plants (WWTP) at the effluent stage and removal from drinking water. Concentrations of drugs in the effluent of WWTP and discarded into rivers may contain higher drug concentrations [39] (<100 ng/L) and present significant unwanted toxicity to aquatic life. On the other hand, concentrations in drinking water accessed from polluted rivers may be lower (<50 ng/L) [39] due to city drinking water treatment plants. However, in dry areas of the world, water may be recycled many times so that drug removal is particularly important in these regions. Removal of drugs at WWTP is generally quite effective and occurs in the activated sludge phase of treatment [14,15,39]. However, two drugs in particular, estradiol and diclofenac, are heavily used and discarded into the sewerage systems and are poorly degraded in the activated sludge [14,15,16,17] and have been added to the European list of drugs of environmental concern [40]. Whilst effluent treatment using ozonation, activated carbon or ultrafiltration (reverse osmosis) are quite effective in removing these drugs [41,42,43,44], these methods are unlikely to be adopted due to the high cost of retrofitting and operation (power) of the WWTP [40]. Furthermore, ultrafiltration or activated carbon treatment methods also have problems with fouling, short lifetimes and environmental disposal [42,44,45,46].

The drug estradiol bound strongly to 0.2 μm filter membranes containing integrated hydrophobically modified CNC. Using higher loadings of CTAB-modified CNC, almost 100% binding of estradiol was achieved on small scale filters (Figure 4a) whereas the drug bound less strongly to covalently modified DDSA-CNC (approx. 70–50%) (Figure 6a,b). Using a commercial over-the-counter household filtration system the DDSA-CNC integrated cartridge (0.2 μm) adsorbed approximately 70% of the estradiol on the first run but maintained a 50% extraction level after that for multiple runs (Figure 7). Although these data do not show a consistent full capture of the drug, they do show the excellent potential of CNC based systems like this to clear estradiol from water. It is unlikely that non-covalently modified CTAB-CNC would be used in household cartridges before a tap outlet as there would be a perceived risk of CTAB eluting into the drinking water. However, in waste water treatment minor amounts of CTAB eluting from membranes would not be problematic due to dilution and because a similar ammonium product [47] is already used in many water treatment systems for pollutant removal.

The charged drug diclofenac was found to bind well to PDMC-CNC integrated membranes both in a small-scale system (approx. 90% binding) or the commercial household filter (a robust 40–50% binding) but, similar to estradiol, it was not possible to achieve 100% efficiencies (Figure 8 and Figure 9). The increased use (and sewage disposal) globally of diclofenac and the poor breakdown by the activated sludge of WWTP has heightened environmental concerns about this drug [1,39,48]. These initial data clearly demonstrate the ability of cationically modified CNC to remove charged species like diclofenac.

In a review of cellulosic material use in waste water treatment technologies, Carpenter et al. [38], pointed out that CNC may provide a support for other active adsorbing particles or may be sorbed onto an existing membrane structure. Furthermore, CNC is easily carboxylated or modified for subsequent covalent modification and hydrophobically trapped species can often be removed by solvent washing with preferable sustainable consequences. All of these features of CNC for waste water treatment have been reinforced in these studies. It should be remembered that nearly all advanced drug elimination procedures like activated carbon traps or ozonation do not fully remove drugs from water [42,43,44]. Drug trapping by ultrafiltration is more efficient but does not achieve 100% removal efficiency and membrane fouling is a persistent problem [42,45,46]. Yangali-Quintanilla et al. [45] showed that a 300 kPA pressure ultrafiltration system allowed for approximately 90% of estradiol to be removed but fouling strongly affects removal efficiencies for numerous drugs. Carbon trapping is also associated with fouling problems [41,42] requiring repeated expensive replacement. Clearly these high-cost, advanced systems are unlikely to be used in most budget-conscious countries. The simplicity and sustainable nature of the use of modified CNC provided in these studies may offer an additional tool for removing drugs from waste water and particularly from drinking water where existing cartridge filters may be easily modified and fouling by other contaminants is less likely than in waste water.

Non-covalently modified cationic CTAC-CNC provided a powerful flocculating agent for clay as shown in Figure 10. Clays are tiny disc-shaped solids with particularly complex charged species often with arrays of both positive and negative charges on the outer rim surface of the disc and a negative charge on the disc faces. The negative charge on clays often increases with higher pH so that at pH 8, as in the oil sands ponds, this use of cationic flocculating species is an attractive strategy [19]. However, numerous amphoteric or zwitterionic cellulosic polymers have been studied with moderate success as flocculating agents [49]. These agents may allow flocculation by bridging or charge neutralization (easiest) processes. In the review paper by Koshani et al. [49], the simplest and most commonly used flocculating method has been to use cationic quaternary ammonium surface modification of many types of cellulose, microcellulose and nanocellulose. Cationically modified cellulose in macro form has been shown to flocculate numerous pollutants including clays (Table 2 in [49]). Quaternary ammonium cationic modification of hairy nanofibers of cellulose (HNC) or cellulose nanofibers results in good flocculants for clays often in the presence of calcium species (Table 3 in [49]). The excellent flocculation of clay observed by Campano et al. [50] was achieved by using hairy cellulose nanofibers where only the end groups on the chains were cationially charged, allowing the long chains to bind to the clay disc faces and bridge and neutralize the negative surface charge. In a similar manner, it is proposed that that the strong positive charge of the CTAC-modified CNC may interact directly with the negative surface charge of the clay causing the observed flocculation.

Naphthenic acids present as pollutants in oil sand tailings ponds are a series of aliphatic and alicyclic carboxylic acids with a pKa in the 5–6 range so at the pH of oil sands (approximately 8) these compounds are generally negatively charged but also vary significantly in their degree of hydrophobicity [25,27,29]. In these studies, we used a more aromatic naphthyloxy butanoic acid and a more complex mixed aliphatic/aromatic compound often used in oil sands naphthenic acid toxicity studies, butyl phenyl butanoic acid [51,52]. Naphthenic acids pollutants are present in oil sands tailings ponds at up to 120 µg/mL concentrations [53,54] and so in these binding studies the naphthenic acids were used at similar concentrations. Both naphthyloxy butanoic acid and butyl phenyl butanoic acid bound strongly (approximately 90%) to the hydrophobically modified CTAB-CNC, as shown in Figure 11. There was a lower level of binding (approx. 40%) to positively charged CTAC-CNC. There are few reports of cellulose based products to clean up naphthenic acid extracts but Frank et al. [28] used a weak anion exchanger with DEAE-modified cellulose which allowed for approximately 41% efficiency of extraction.

Other workers report the use of flocculation to bind and separate naphthenic acids using the positively charged PDMC (poly(diallyldimethylammonium chloride) (PDMC is the same as the covalently cationic species used in the water purification part of this study) with 37% efficiencies (reviewed in [27]). More recently ozonation of naphthenic acids has been shown to produce oxidized naphthenic species which are more readily biodegraded [27,55] and interestingly may be flocculated using PDMC at higher efficiencies (86%) than observed for native compounds. Ultrafitration methods are quite effective in the laboratory for removing naphthenic acids from water but in the real world rapid fouling of the membranes makes this method unviable [27]. Future studies in this laboratory will focus on synergistic extraction methods whereby CNC-CTAC flocculated clays may coflocculate bound CNC-CTAC or CTAB-CNC naphthenic acid complexes. It is likely that generally lower efficiency water purification steps may be used in combination or in sequence to achieve satisfactory naphthenic acid removal [55]. The inclusion of such steps will strongly depend on budgetary considerations and environmental compatibility such as clean up, disposal or biodegradation. Clearly, nanocrystalline cellulose fits both budgetary and environmental requirements.

## 6. Conclusions

In this study, it has been shown that CNC may be easily surface modified to produce positively charged or hydrophobic surfaces. These modified nanofibres may be integrated into existing 0.2 μm commercial filters using low-pressure filtration and may adsorb either negatively charged drugs like diclofenac or hydrophobic drugs like estradiol. Furthermore, the adsorption of cationic or hydrophobic chemicals onto the surface of CNC provides materials that may be simply mixed with clay suspensions or naphthenic acid solutions to either flocculate clays or bind naphthenic acids. Similar to almost all remediation systems for cleaning water, these CNC-based systems are not 100% effective but may offer a complementary method to other clean-up procedures for a cumulative cleaning effect.

## Figures and Tables

**Figure 1 polymers-13-03899-f001:**
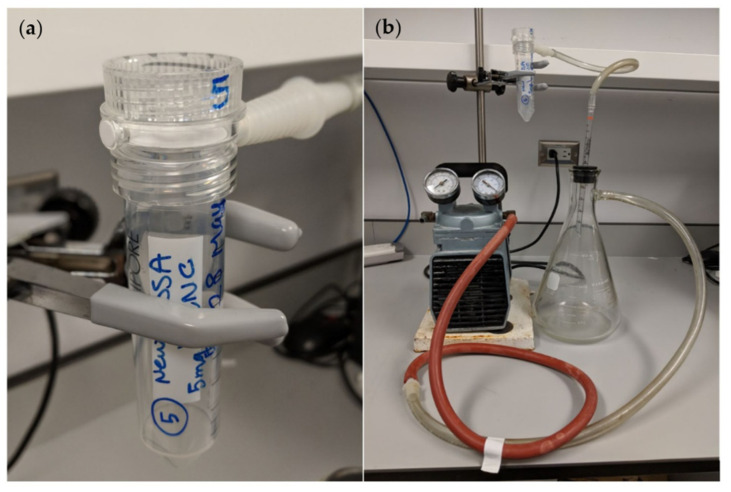
(**a**) Millipore Steriflip filter (0.22 µm). (**b**) Setup for the small-scale filtration.

**Figure 2 polymers-13-03899-f002:**
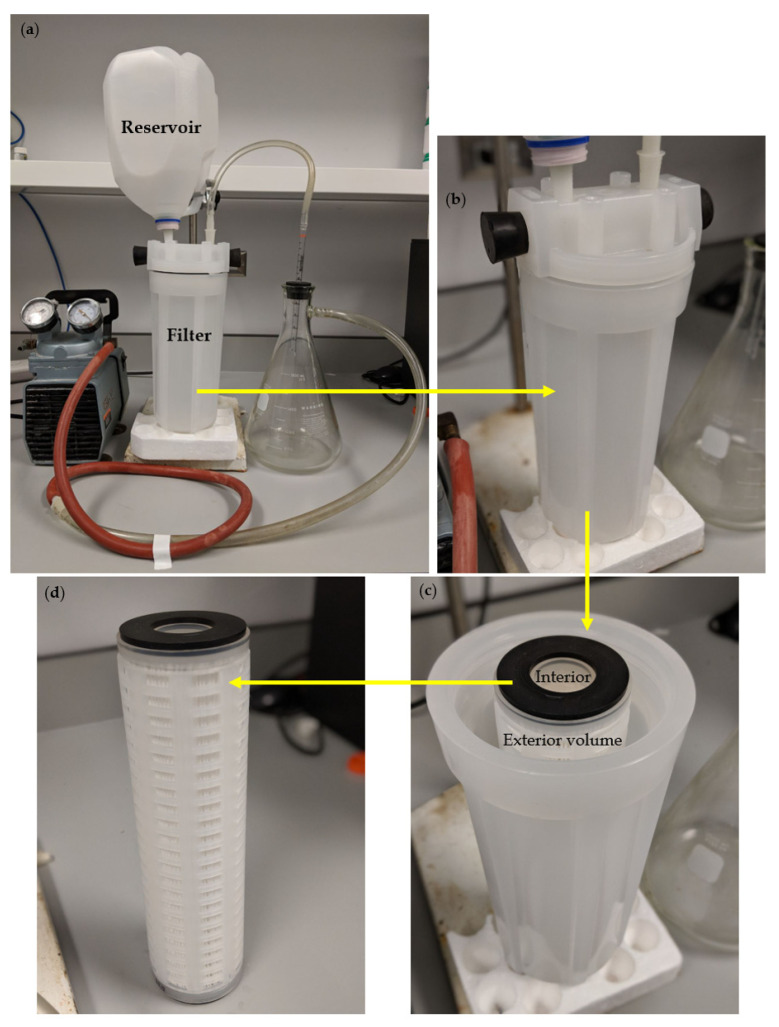
The setup for the large-scale (commercial) filter. (**a**): full system, (**b**): sealed filter chamber, (**c**): inner chamber and (**d**) 0.2 µm filter.

**Figure 3 polymers-13-03899-f003:**
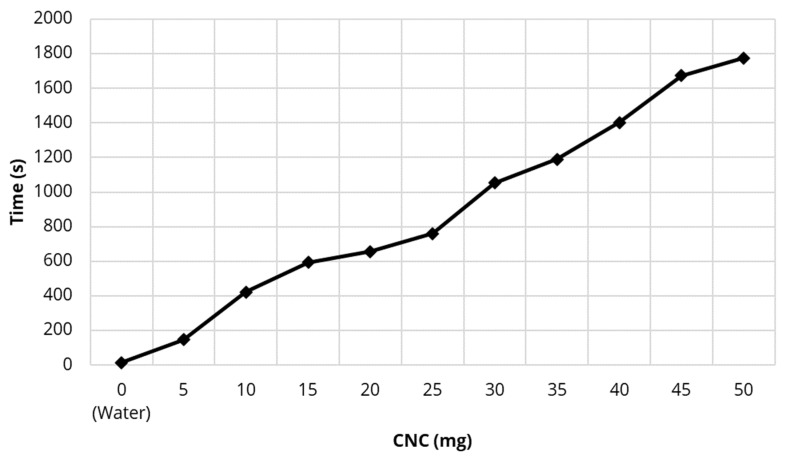
Effect of filtering increasing amounts of nanocrystalline cellulose (CNC) onto the Millipore small-scale filter on the time for the suspension to filter.

**Figure 4 polymers-13-03899-f004:**
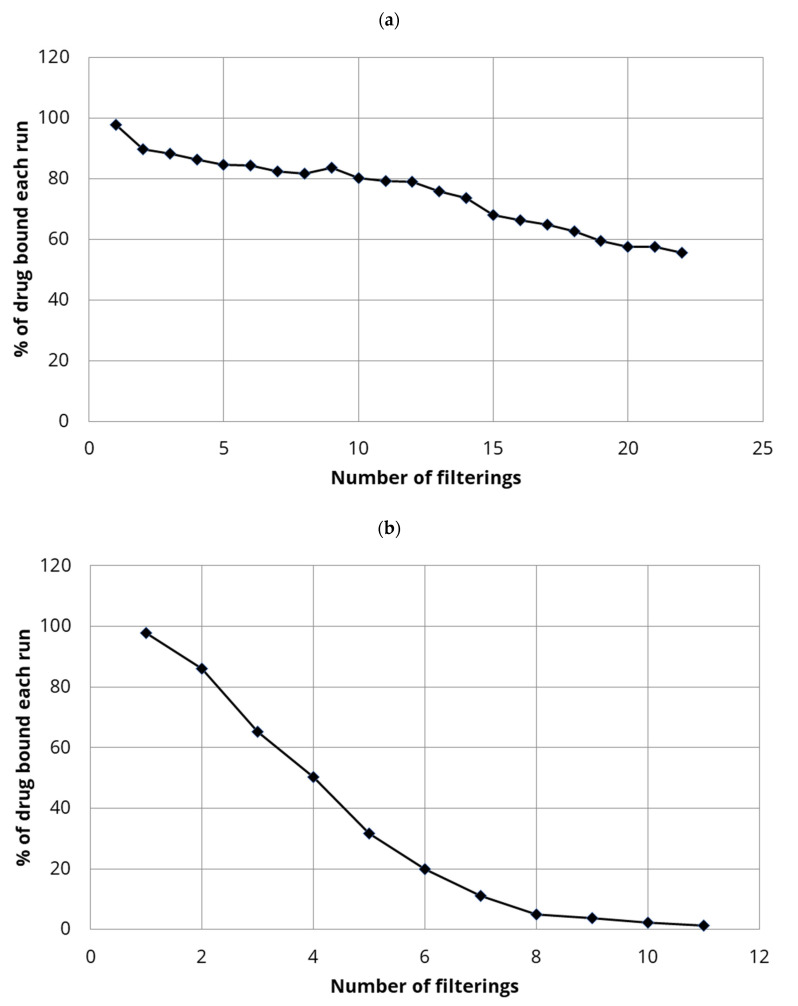
(**a**). Binding of estradiol (repeated 1 mL of 2 µg/mL) to hydrophobic CNC (CTAB-CNC) (25 mg of CTAB-CNC integrated into small-scale filter). (**b**). Binding of estradiol (repeated 5 mL of 2 µg/mL) to CTAB-CNC (5 mg of CTAB-CNC integrated into small-scale filter).

**Figure 5 polymers-13-03899-f005:**
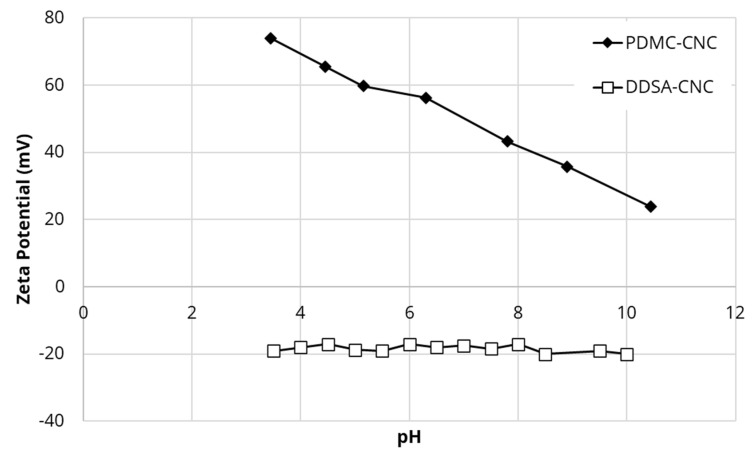
Zeta potential of covalently modified CNC as a function of pH.

**Figure 6 polymers-13-03899-f006:**
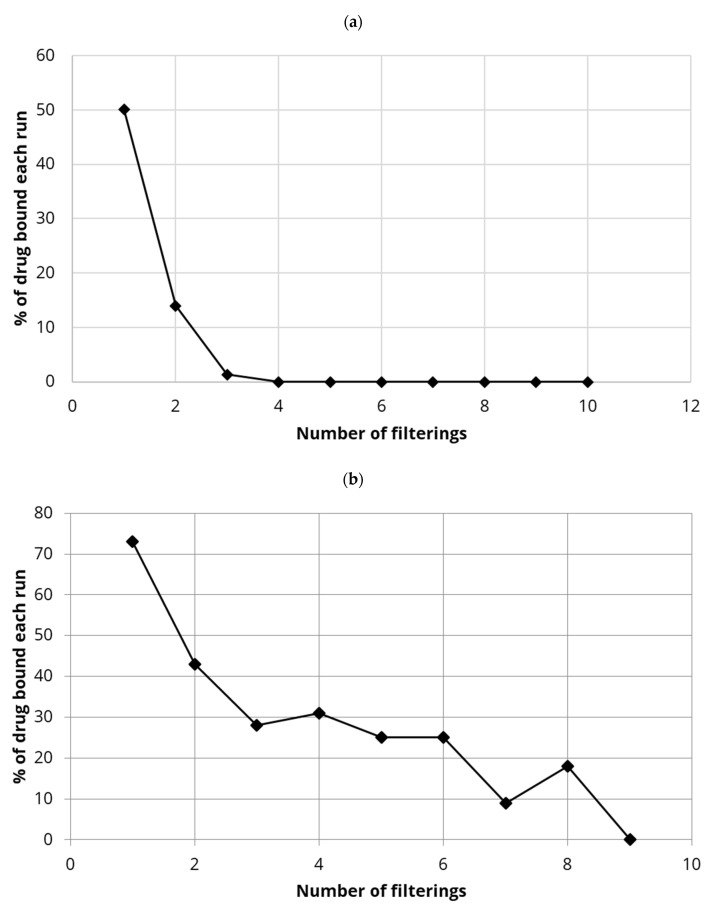
(**a**). Binding percentage of estradiol (repeated volumes of 5 mL of 2 µg/mL) for DDSA-CNC (5 mg integrated into the small-scale filter). (**b**). Binding percentage of estradiol (repeated volumes of 5 mL of 2 µg/mL) for DDSA-CNC (10 mg integrated into the small-scale filter.

**Figure 7 polymers-13-03899-f007:**
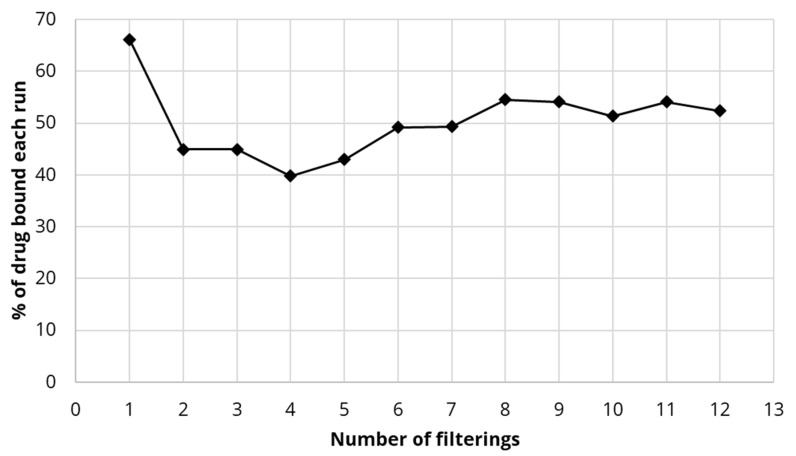
Binding percentage of estradiol (repeated volumes of 2 L of 2 µg/mL) for DDSA-CNC (2 g) integrated into commercial filter.

**Figure 8 polymers-13-03899-f008:**
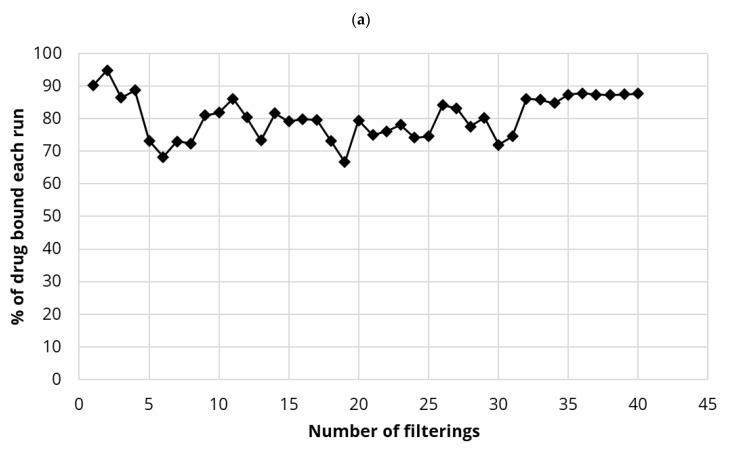
(**a**). Binding percentage of diclofenac (repeated volumes of 5 mL of 2 µg/mL) for polydiallyl dimethyl ammonium chloride-CNC (PDMC-CNC) (5 mg integrated small-scale filter. (**b**). Binding percentage of diclofenac (repeated volumes of 5 mL of 2 µg/mL) for PDMC-CNC (20 mg) integrated small-scale filter.

**Figure 9 polymers-13-03899-f009:**
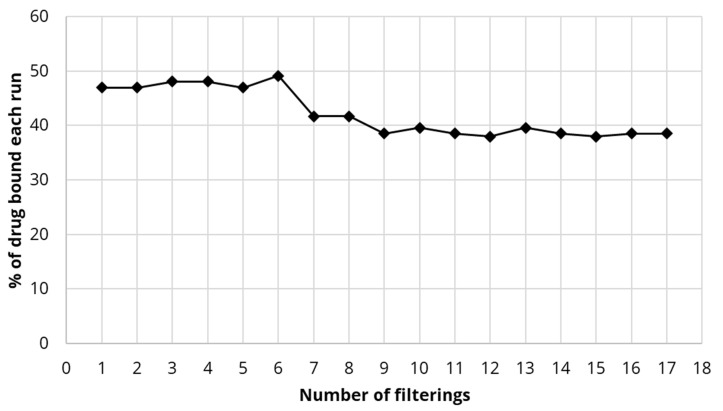
Binding percentage of diclofenac (repeated volumes of 2 L of 2 µg/mL) for the PDMC-CNC (2 g) integrated commercial filter.

**Figure 10 polymers-13-03899-f010:**
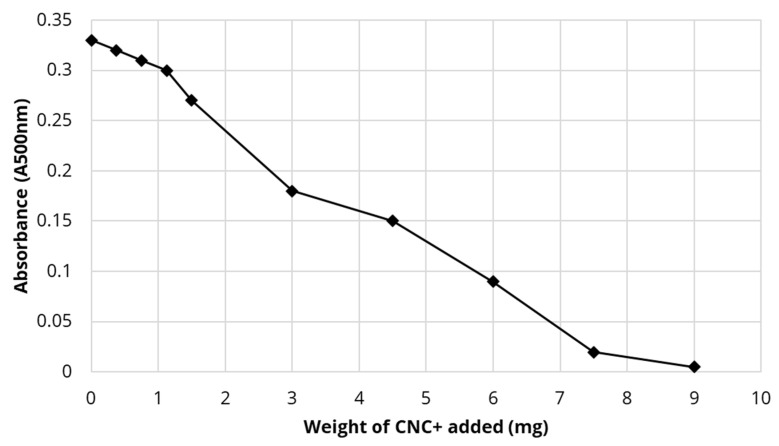
Flocculation of Cloisite clay (20 mg) by CTAC-CNC. Turbidity absorbance at 500 nm after mild centrifugation.

**Figure 11 polymers-13-03899-f011:**
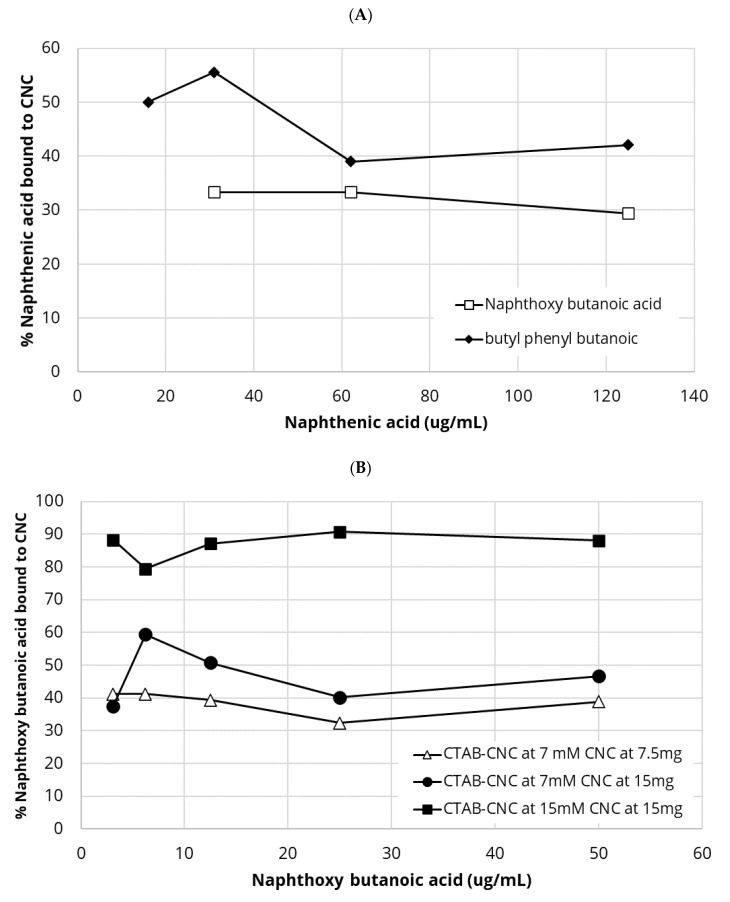
(**A**) Binding of butanoic acids to cationic CNC. CTAC-CNC. (**B**) Binding of naphthoxy butanoic acid to CTAB-CNC. (**C**) Binding of butyl phenyl butanoic acid to CTAB-CNC.

**Table 1 polymers-13-03899-t001:** Zeta potential of CNC and clay.

Sample Materials	Zeta Potential (mV)
Cloisite clay	−40 mV
Nanocrystalline cellulose (CNC)	−50 mV
Cationic CNC (CTAC-CNC)	+41 mV
CTAB-CNC (15 mM CTAB)	−20 mV

## Data Availability

Not applicable.

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
