# Peer review of "The Use of Surface-Modified Nanocrystalline Cellulose Integrated Membranes to Remove Drugs from Waste Water and as Polymers to Clean Oil Sands Tailings Ponds"

_polymers, 2021, doi:10.3390/polym13223899_

Round 1
Reviewer 1 Report
The manuscript polymers-1436693 presents the potential of modified CNC (with polydiallyl dimethyl ammonium chloride, PDMC, or dioctenyldimethyl succinic anhydride, DDSA) to be use in wastewater treatments, by them integration into membrane filters, in order to remove estradiol or diclofenac. Moreover, the effect of different modified CNC (with 3- chloro hydroxy propyl trimethyl ammonium chloride, CTAC or cetyl trimethyl ammonium bromide, CTAB) to flocculate clays and bind naphthenic acids, was studied.
The manuscript still needs a lot of work in order to be good for publication, so I recommend the publication in Polymers journal, after major revisions:
Abstract
- L. 14, L.15-16: Please explain in text how CNC was modified, for both cases!
Introduction
- Literature studies on pharmaceutical actives compounds (drugs) and hormones retention systems are not reviewed! Please add the recent advances that have been achieved in wastewater treatment processes, with the corresponding references!
- There are no examples related to the use of CNC for removal of drugs or hormones from waste water, even if this is the subject of this study! Please add this information and include the corresponding references!!!
- Even if the references [4], [5] and [6] refer to diclofenac and estrogen, there is no information about the use of cellulose to remove these compounds from waste water!
- There is no information about CNC! The only paragraph from entire Introduction section is L. 91-93! Please mention the extraction methods for CNC and their important characteristics and give the recent reviews related to this subject!
- L. 70-80: This paragraph is of no importance for the manuscript and must be removed!
- L. 86-88: There are only few data related naphtenic acids and the problems caused by them! Please expand this paragraph with more information about the nature and the types of naphtenic acids, the adverse effects on the environment, the toxicity of them, and the techniques to isolate and treat them, with corresponding recent references.
- Please add information and references related to the use of CNC to flocculate clay!!!
- L. 88-90, L. 96-100: There are two aims of the manuscript presented separately! The authors begin with the second aim of the paper, present some theoretical information about CNC (L. 91-96) and then present the general aim od this study! Please revise the entire paragraph L. 88-100, and present a single, one clear aim of this study!
- L. 96: What is the reason to mention the use of CNC as controlled drug release systems, if this is not the subject of this study? Please remove this information!
- Please explain what is new about this study, in report with the studies already existent in literature!
- The entire Introduction section must be restructured and a lot of information related to the present study must be added!
Materials and Methods
- There is no paragraph related to chemical substances used in this study! Please add information (characteristics, company, etc.) for all the substances used in this study in a separately section, MATERIALS! Moreover, there is no information about the CNC used in this study, which is a very important information! Please carefully complete this paragraph with all the requested information!
- Please present separately in Method section all the methods used in this study and the corresponding equipment (name, company, characteristics) used for the characterization of the prepared materials (such as, optical density, zeta potential, flocculation, turbidity, etc.), in order to be clear how was done each method of characterization!
DDSA-CNC synthesis
- L. 116-118: What is the temperature of reaction?
Preparation of cationic CNC or hydrophobic CNC (non covalent methods).
- Please delete the point from title!
- There is an ambiguous nomenclature for the materials prepared in this section!!! There is an inconsistency in the name of the sample CTAB-CNC, which appears also as CNC-CTAB! Moreover, both notations appear in L. 187-188!!!
- Related to CNC-CTAC (L. 140), this appears only at the end of manuscript (L. 486-487) and nowhere else in the manuscript!! Please use it in the entire manuscript!
- Please revise the entire terminology and use CTAC-CNC and CTAB-CNC throughout the manuscript, as the other samples are noted!
- Please use the codes of the samples and avoid to use CNC+, cationic CNC, hydrophobic CNC, in order to eliminate the ambiguous notations within the manuscript, that I will mention forward!
Drug or contaminant analysis.
- The same observation related to the dot from title!
- L. 143: “Diclofenac was measured using UV/VIS spectrophotometric methods”?? Maybe “the concentration of diclofenac”! There is no information about UV/VIS spectrophotometer! This information must be added separately in Methods section!!!
Filtration.
- Please delete all the points after the titles in the whole manuscript!
- Figure 1a and 1b are presented as two independent figures, therefore it would be better to change them as: Figure 1a to Figure 1 and Figure 1b to Figure 2, in order to evidence the use of two different systems, as the authors intended!!
Drug binding studies.
- L. 180: “The PDMC modified CNC”?? There is already a code for this sample, such as PDMC-CNC, as it is mentioned at L. 129! Please use the codes within entire manuscript!!!
- L. 182-183: “the amount of drug in the filtered liquid (unbound fraction) was analyzed and discarded”? How was analyzed?
- Please revise the nomenclature: CNC-CTAB and CTAB-CNC!! If the authors want to refer to two different experiments by inversing the notation, this it is not acceptable and must use different codes!
- L. 192: “DDSA modified CNC” must be DDSA-CNC! Please revise it in the entire manuscript!
Filtration methods:
- The same observation with “:” from the title!
- L. 217: “to filter numerous repeat 5 mg weights of CNC through the filter”?? Please revise the explanation!
Binding of Beta-estradiol to CNC-CTAB integrated small scale membranes
- L. 236-238: “However, using 25mg of CNC-CTAB after 20 filtrations of 236 1 mL (total of 40ug of drug) there was still 60% binding whereas for 5 mg CNC-CTAB 237 filters after 4 filtrations of 5 mL (total 40ug of drug) there was 50% binding.”! Please revise the paragraph and explain what “binding” refers!
- L. 241-246: Figure 3 and Figure 4 must be integrated in the same figure with notations Figure 3a and Figure 3b, due to the fact that present the same effect – the influence of CNC-CTAB concentration on the drug binding!
- There is again a problem of sample notations! Why Beta-estradiol if until now, in the manuscript appear estradiol? In addition, here CNC is before CTAB!! Please revise the terminology and maintain it the same throughout the work!
Zeta potential of covalently modified CNC.
- Please explain the surface charge modifications as a function of the values obtained for zeta potential and how these influence the drug and hormone binding!
- Why is important this section for the study? Did the authors use changes in pH in the proposed experiments? Please give an explanation!
- At the caption of Figure 5 there is mentioned that the zeta potential is established for “covalently modified CNC”, but in the legend appear both, PDMC (cationic) and DDSA (covalently)! Please make the adequate corrections and use the codes already used in the manuscript!!!
- Moreover, there is a problem of nomenclature in the legend of Figure 5! In my opinion, must be PDMC-CNC and DDSA-CNC! Please revise the legend!
Binding of Beta-estradiol to DDSA-CNC integrated small scale membranes
- Please revise the y-axis of Figure 6 and delete the decimals!
- In the case of Figure 6 and 7, in my opinion would be better to change them in Figure 6a and 6b (to be used the correct number of figure - this is just an example), because are related to the same experiment!
Binding of Beta-estradiol to DDSA-CNC integrated large commercial filter membranes
- Please revise the y-axis of Figure 8 and delete the decimals!
- L. 269: Please mention in text not only “DDSA-CNC”, but also the weight of sample (mg), concentration of the solution, volume of solution, etc.!!
- Figure 8 has an unusual trend! Please explain in text why the binding of estradiol decreases until the 4th run and then increase!
Binding of diclofenac to PDMC integrated small scale membranes.
- Please revise the y-axis of Figure 9 and 10 and delete the decimals!
- Please change the name of Figure 9 and 10 in Figure 9a and 9b (to be used the correct number of figure) and couples the figures in a single figure, as was required in previous cases (Figure 3a and 3b and Figure 6a and 6b)!!
Oil sands treatments: clay flocculation
- In Table 1, the first column represents the “Samples” notation and not “Zeta potential of materials”!!! The second column must be “Zeta potential (mV)”!!! Please make the adequate corrections!
Binding of naphthenic acids to either CNC+ or CTAB- modified CNC.
- The same observation related to nomenclature of the samples is related to the title of this section. Please try to use a unique and clear nomenclature in the entire manuscript! Please revise the title of this section!
- “binding of naphthenic acids to CNC+”??? Please revise it!
- Moreover, the authors can’t discuss about two different experiments with the generally “naphthenic acids”, they must explain separately each experiments!!
- There is no codes for the samples which bind to (i) butyl phenyl butanoic acid and (ii) napthobenzoic acid!! Please add independent codes in order to be easy to follow the modifications!
- In Figure 13, even if the caption of the figure mentioned different experiments, from the legend of Figure 13b and Figure 13c there is no differences and thus the observations become confusing. Please make independent codes for each experiment and make the adequate corrections in Figure 13.
Discussions
- L. 355-357: “Because of the ability to bind drugs, CNC has also been used in formulations for drug delivery underlining the potential for the material to sequester drugs from polluted waters”?? This is an ambiguous sentence! Please revise it!
- In Discussion section there are the same explanations for the experiments as it was already written in Results section!
- There is no addition explanations related to the differences in drug binding when it was used: (i) 5 mg or 20 mg PDMC-CNC, or for the cases of (ii) 5 mg DDSA-CNC and 10 mg DDSA-CNC, or (iii) 5 mg CNC-CTAB and 25 mg CNC-CTAB!!! The authors present only the trends or the final percent of binding, but nothing about the phenomena.
- Moreover, this section is almost only literature data!! The paragraphs related exactly to the study are found only at L. 369-379 (where are presented the zeta potential values of the samples, the same explanation as in Results section), L. 397-403 (data for DDSA-CNC and for CNC-CTAB) and L. 411-414 (for PDMC-CNC).
- Please explain why appear these differences in binding processes when it is used different amounts of samples!
- Please explain why the authors used different quantities of samples for different studies and they didn’t maintain the same amounts for all three experiments!
- Please explain why the drug bound is less strongly for the case of covalently modified DDSA-CNC sample in comparation with the case of non-covalently modified samples!
- Please make a discussion related to the drug binding for the case of covalently and cationic modified samples and explain these processes!!!
- Please revise the entire paragraph and make the requested modifications!
Conclusions
- Please add a Conclusion section where must be emphasized all the achievements obtained in this study!
Author Response
Please see attachment.
Thanks for doing such a detailed review. I must apologise for the poor presentation of the initial manuscript.

Reviewer 2 Report
- Thank you for submitting your paper. The work done here draws attention to a significant subject in condition monitoring using surface modified nanocrystalline cellulose with membranes for environmental applications. I have found the paper to be interesting. However, several issues need to be addressed properly before the paper is being considered for publication. My comments including major and minor concerns are given below:
- Please consider reviewing the abstract and highlight the novelty, major findings and conclusions. I suggest reorganizing the abstract, highlighting the novelties introduced. For example, line 22-26 are somewhat generic, it is not clear what exactly was concluded from this work, it should contain answers to the following questions:
- What problem was studied and why is it important?
- What methods were used?
- What conclusions can be drawn from the results?
- What is the novelty of the work and where does it go beyond previous efforts in the literature?
- The abstract should be one clear paragraph instead of three, please combine all of them together.
- Just before the last paragraph in the introduction, the authors should answer the following question: What is the research gap did you find from the previous researchers in your field? Mention it properly. It will improve the strength of the article.
- The introduction part is short and should be expanded upon. The literature review is basic and generic about how processes occur, it does not provide the readers with a clear understand of the problem in hand. Please use more in-depth critical review. Also, please cite papers related to your work from mdpi journals. Please provide reports on past studies similar to your work or closely related to it, mention what they did and what were their main findings. Then explain how does your current work brings new knowledge and difference to the field.
- Please avoid bulk citations such as in line 88 references for a single sentence.
- Please change “Materials” to “Materials and Methods”
- Materials and methods section is comprehensive and clear, however, images and graphs of equipment used, samples fabricated, and tests implemented with details on those images should be provided, this is an experimental study, and it is important to give sufficient information to the readers about the work done here.
- Please avoid using we, us or our, please check this issue everywhere in the manuscript.
- Please add more references from mdpi journals that are related to this work.
- Please combine figures 3/4 and 7/8 and 9/10 in one figure, the data is similar and can be all plotted in one graph for each of comparison and analysis.
- Lines 349-361 please move them to the introduction; they don’t belong to the discussion.
- I strongly advise the authors to have one section called “Results and Discussion” instead of two separate sections to better correlate the results in the graphs with analytical discussion directly.
- Please combine smaller paragraphs into larger ones. For example, 180-195 combine into one larger paragraph. Please check this issue elsewhere in the manuscript.
- Good discussion, all findings were explained and correlated to open literature recommended to merge with results for more clarity.
- The authors are encouraged to include a more detailed results and discussion section and critically discuss the observations from this investigation with existing literature.
- Conclusion is missing!
- The article is not adequately organised, lacks basic formatting of polymers manuscript format. Please make sure to follow the guidelines and adequately structure the content of this work.
Round 2
Reviewer 1 Report
The manuscript polymers-1436693 has been improved over the previous version and therefore, I recommend the publication of this work in its present form.
Author Response
thanks for the detailed review and for making this a much better manuscript
Reviewer 2 Report
All questions answered, but please add (a) and (b) to all figures with multiple images for clarity
Author Response
thanks for reviewing and making this a better manuscript. we have added the (a) etc. to the figures.